# Safety of a Sustainably Produced, Bioengineered, Nature-Identical Salidroside Compound

**DOI:** 10.3390/nu14112330

**Published:** 2022-06-01

**Authors:** Philip G. Kasprzyk, Christopher Vickery, Mingli Ye, Magdalena Sewastianik, Wei Gong, Sheng Ding, Margitta Dziwenka, Amy Mozingo, Kaiti Valm, Holly Schachner, Jing-Ke Weng

**Affiliations:** 1DoubleRainbow Biosciences Inc., Lexington, MA 02421, USA; chris.vickery@doublerainbowbio.com (C.V.); mingli.ye@doublerainbowbio.com (M.Y.); sewastianik@doublerainbowbio.com (M.S.); weigong01@gmail.com (W.G.); sheng.ding@doublerainbowbio.com (S.D.); holly.schachner@doublerainbowbio.com (H.S.); jingke.weng@doublerainbowbio.com (J.-K.W.); 2GRAS Associates—Nutrasource Pharmaceutical and Nutraceutical Services, Guelph, ON N1G 0B4, Canada; dziwenka@gras-associates.com (M.D.); amozingo@gras-associates.com (A.M.); kvalm@nutrasource.ca (K.V.); 3Whitehead Institute for Biomedical Research, Cambridge, MA 02142, USA; 4Department of Biology, Massachusetts Institute of Technology, Cambridge, MA 02139, USA

**Keywords:** salidroside, toxicity, safety, NOAEL, *Rhodiola*

## Abstract

Bioactive phytochemicals such as salidroside have been studied to understand the beneficial effects of *Rhodiola rosea*, an herbaceous plant used in traditional medicine to increase energy and treat a variety of health issues. However, Rhodiola plants are often slow-growing, and many are endangered in their native habitats. Thus, there is a need for safe, alternative supplies of key phytochemicals from Rhodiola. The salidroside subject of this safety study is a synthetic biology product from fermentation of a bioengineered *E. coli* that produces salidroside. Here, we present comprehensive test results that support the safety of salidroside manufactured via a patented sustainable bioengineering manufacturing process. In vitro bacterial reverse mutation assays with the bioengineered salidroside show no mutagenicity in any of the concentrations tested. In vivo toxicity studies in rats show no adverse effects from the salidroside product. Based on the results of these studies, we conclude that the bioengineered salidroside discussed here is not genotoxic and demonstrates a no-observed-adverse-effect level (NOAEL) at least 2000 mg/kg bw/day in male and female Sprague–Dawley rats. This study supports that the salidroside compound produced using bioengineered *E. coli* is a viable alternative to salidroside produced from harvested Rhodiola plants for use as a dietary supplement, food ingredient, or potentially as a pharmaceutical product.

## 1. Introduction

*Rhodiola rosea* is an herbaceous plant used in traditional medicine to increase energy and treat a variety of health issues. Bioactive phytochemicals, such as the 2-phenylethanol derivative salidroside, have been studied to understand the beneficial effects of this plant [1]. However, sourcing of *Rhodiola* spp. plants to supply Rhodiola extract is threatened, and there is a need for safe, sustainable, alternative supplies of key phytochemicals.

*Rhodiola rosea* grows at high altitudes in Europe and Asia, while other *Rhodiola* spp. are circumpolar [1,2]. In areas such as the Xinjiang region of China, *Rhodiola rosea* is harvested by hand in difficult terrain between late June and August and makes up the majority of the harvesters’ annual income [3]. Although harvest management plans exist for wild native medicinal plants, they are not pragmatic and are hardly implemented [4]. In addition to threats from climate change [5], unregulated harvest confers compounding threats to the survival of *Rhodiola rosea*, for example by increasing the distance between male and female plants, making reproduction more difficult [6].

As early as 2012, research published in the Scientific World Journal noted *Rhodiola rosea*’s status as an endangered species, and several European and Asian countries, including Bulgaria, the Czech Republic, Germany, and Mongolia have added it to their threatened species registries [7]. In 2021, experts from the US, South Africa, Australia, and Germany called for changes to the global *Rhodiola rosea* supply chain, as increasing global demand threatens wild Rhodiola sources [8,9].

Cultivation of *Rhodiola rosea* and other *Rhodiola* spp. is one possible method to take the pressure off wild populations [6]. However, in experiments conducted in the former Soviet Union, Poland, Sweden, and Finland, cultivation success was challenged by the long maturation time from planting to harvest (~5 years) and labor-intensive harvesting and processing [10]. Although this strategy could be an important economic lever to preserve wild Rhodiola populations, it has its own limitations and may not meet consumer demand in the short term. In addition, adulteration with other *Rhodiola* spp. or other herbal products occurs in the supply chain [1,11,12] and could increase for *Rhodiola rosea* if supply is reduced or becomes too expensive.

In recent years, synthetic biology has emerged as a new strategy to produce bioactive phytochemicals or “nature-identical” chemicals in genetically engineered microbial hosts. Salidroside is considered one of the primary bioactive compounds in *Rhodiola rosea* [13,14]. The salidroside subject of this safety study is a synthetic biology product from fermentation of a bioengineered *E. coli* that produces salidroside. The intended use of salidroside is as a dietary supplement with indications of use comparable to those of salidroside from Rhodiola extract. Yet the USA Food and Drug Administration does not consider “nature-identical” phytochemicals from synthetic biological processes to qualify under the definition of a new dietary ingredient [15]. As such, the regulatory pathway required to establish safety is the Generally Recognized as Safe (GRAS) pathway. The GRAS pathway requires that pivotal information supporting the safety of a proposed food ingredient be publicly available and generally recognized as safe by qualified experts.

To demonstrate the safety of the “nature-identical” salidroside, standard assays and studies were conducted. The Bacterial Reverse Mutation (Ames) assay is used to evaluate if a compound can change the genetic sequence in a selection of bacteria as an indication of mutagenicity potential. The single dose toxicity study gives an indication of acute toxicity, and the dose used is often well above the intended use. The 7-day and 28-day repeat dose studies are used to provide an indication of any effects from repeat exposure. These standard tests were conducted to evaluate any potential health risks from consumption of the bioengineered salidroside. Collectively, the results of these studies support the safety of salidroside manufactured via a patented manufacturing process [16].

## 2. Materials and Methods

### 2.1. Test Material

The test article used in the studies was salidroside (white crystalline powder; 99.3% pure, CAS #10338-51-9) and was supplied by LandKind, Lexington, MA, USA, a subsidiary of DoubleRainbow Biosciences Inc. All studies were conducted at Product Safety Labs (PSL, Dayton, NJ 08810) and were compliant with FDA GLP (21 CFR Part 58, 1987). For the in vivo studies, all animals were housed following husbandry methods conforming to the most recent Guide for the Care and Use of Laboratory animals [17] and approved by the local Institutional Animal Care and Use Committee.

### 2.2. Bacterial Reverse Mutation (Ames) Assay

The mutagenicity potential of the test article was evaluated in the Bacterial Reverse Mutation (Ames) Assay in accordance with OECD Guidelines for the Testing of Chemicals, Section 4, Test No. 471 [18]. Four strains of *Salmonella typhimurium* (TA98, TA100, TA1535 and TA1537) and one strain of *Escherichia coli* (WP2 uvrA) were used. The fresh bacterial suspension cultures were prepared in nutrient broth and were in the late exponential phase of growth when used. The salidroside was prepared in sterile water to provide the required concentrations and vortexed just prior to use. The studies were conducted in the presence and absence of a metabolic activation system from male Sprague–Dawley rats that had been induced with phenobarbital and 5,6-benzoflavone (Molecular Toxicology Inc., Boone, NC, USA). The overlay agar and minimal glucose agar plates used in the study were purchased from Molecular Toxicology Inc., Boone, NC, USA. The following positive control substances were included in the study: sodium azide, 9-aminoacridine hydrochloride monohydrate, 2-nitrofluorene, 4-nitroquinoline N-oxide, 2-aminoanthracene, and benzo[a]pyrene. The initial test was conducted as a plate incorporation assay in which the prepared test article solutions, vehicle control, or positive control substance; S9 mix or substitution buffer; bacteria suspension and overlay agar were mixed and poured over the surface of a minimal agar plate. The agar was allowed to gel, then the plates were inverted and incubated at 37 °C until growth was adequate for enumeration. The plates were prepared in triplicate at the following concentrations: 0, 1.58, 5.0, 15.8, 50, 158, 500, 1580 and 5000 µg/plate. A confirmatory assay was conducted as a preincubation test in which the test or control articles, bacteria suspension and S9 or substitution buffer were incubated for approximately 30 min at approximately 37 °C and then mixed with the overlay agar and poured onto the minimal agar plates. The agar was allowed to gel, then the plates were inverted and incubated at 37 °C until growth was adequate for enumeration. The plates were also prepared in triplicate at the same concentrations as the initial test. At the end of the incubation period, the number of colonies per plate were enumerated either manually or with a plate reader, and the mean and standard deviations were calculated. The background lawn was evaluated to determine any toxic effects of the salidroside. For each experimental point, the Mutation Factor (MF) was calculated, and the mutagenic activity of the test material evaluated. The results were considered to show mutagenic potential if there was a substantial increase in the revertant colony counts as compared to concurrent and historical controls, and this result must be dose-related and/or reproducible.

### 2.3. Acute Single Dose Oral Toxicity Study

The acute oral toxicity study was conducted to determine the potential for salidroside to induce toxicity following a single oral dose utilizing the up and down procedures detailed in OECD guideline 425 [19] and in compliance with USA FDA GLP (21 CFR Part 58). The study was conducted in female Sprague–Dawley rats that were group housed with the exception of dose day, when they were single housed following dosing and then returned to group housing. Rats had *ad libitum* access to food and water with the exception of overnight fasting prior to dosing. The diet used was Envigo Teklad Global 16% Protein Rodent diet^®^ 2016. The salidroside was prepared in distilled water just prior to dosing and mixed well prior to use. Based on the expected safety profile, the study was conducted as a limit test at 5000 mg/kg bw given as a single oral dose using a stainless-steel gavage needle. Initially, a single female was dosed and when no mortality was reported, two additional animals were given the same dose. The final volume given was based on the most recent bodyweight taken, and animals were fasted for approximately 3–4 h after dosing. All animals were observed for mortality, gross toxicity, and behavioral changes at least once daily for 14 days following dosing. Bodyweight was recorded prior to dosing and then again on days 7 and 14. All animals were euthanized and underwent a full gross necropsy on day 14.

### 2.4. Seven-Day Oral Range Finding Study

A 7-day oral range finding study was conducted in male and female Sprague–Dawley rats to determine the appropriate doses for the 28-day study. The study was conducted in compliance with OECD Guideline 407, OPPTS 870.3050 and Redbook 2000, IV.C.4 [20,21]. Forty animals were divided into four groups (n = 5/sex/group), which included one control and three salidroside-treated groups at 500, 1000, and 2000 mg/kg bw/day. The rats had *ad libitum* access to food (2016 Certified Envigo Teklad Global Rodent Diet^®^) and water for the duration of the study. The salidroside was prepared in distilled water, which was also used as the control The dose formulations were prepared daily and maintained on a magnetic stir plate during dosing. The individual doses for each rat were calculated from the most recent bodyweight, and animals were dosed with a stainless-steel gavage needle at approximately the same time each day for the duration of dosing. Animals were observed daily by cage-side observations and twice daily for mortality. Detailed hands-on examinations were done weekly on all animals. Individual animal bodyweights were recorded on day 1 prior to dosing and then weekly thereafter. A final fasted bodyweight was recorded prior to euthanasia. Bodyweight gain was calculated for selected intervals. Individual animal food consumption was measured weekly to coincide with the bodyweight recordings. On the final day, animals were euthanized by carbon dioxide asphyxiation, and a gross necropsy was performed on each animal.

### 2.5. Twenty-Eight-Day Repeat Dose Study

The potential subchronic toxicity of salidroside was evaluated in a 28-day repeat dose oral toxicity study in male and female Sprague–Dawley rats. The study was conducted in compliance with OECD Guideline 407, OPPTS 870.3050 and Redbook 2000, IV.C.4 [20,21,22]. Eighty rats were divided into 4 groups (n = 10/sex/group); control, 500, 1000, and 2000 mg/kg bw/day. These dose levels were selected based on the results of other in vivo studies conducted with salidroside and the expected human intake. Distilled water was used as the control article. Animals were group-housed by sex with no more than two per cage and had *ad libitum* access to water and food (2016 Certified Envigo Teklad Global Rodent Diet^®^) with the exception of an overnight fast prior to euthanasia at the end of the study. The animals were acclimated for 14 days prior to dosing. The test article was prepared in distilled water, and fresh formulations were prepared daily. Samples of the dose formulations were taken at the beginning, approximately the middle, and then at the end of the study for verification of dose concentration. Samples from each concentration and the control were taken at each timepoint. Stability of the neat test article was also evaluated over the course of the study. The formulations were maintained on a magnetic stir plate during dosing. Individual animal doses were calculated based on the most recent weekly bodyweights. Doses were given by gavage using a stainless-steel ball-tipped gavage needle and syringe and given 7 days/week for at least 28 days. Doses were given at approximately the same time each day. Cage side observations were conducted on all animals once daily, and all animals were observed twice daily for mortality. Detailed clinical hands-on examinations were conducted on day 1 prior to dosing and then weekly thereafter. A functional observational battery (FOB) was performed on all animals on day 22 of the study, which included forelimb and hindlimb grip strength and foot splay measurements. Motor activity was evaluated on all animals during the same time period as the FOB evaluations. All animals were weighed on study day 1 prior to dosing, weekly thereafter, and then prior to euthanasia following an overnight fast. Bodyweight gain was calculated for specific intervals and then for the study overall. Individual food consumption was determined to coincide with the bodyweight measurements. At the end of the study, blood was collected for hematology, clinical chemistry, coagulation, and thyroid hormone analysis. Hematology parameters evaluated included hematocrit, hemoglobin concentration, mean corpuscular hemoglobin, mean corpuscular volume, platelet count, red blood cell count, red blood cell distribution width, reticulocyte count, total white blood cell, and differential leukocyte count. Mean corpuscular hemoglobin concentration was calculated. Clinical chemistry parameters evaluated included albumin, alkaline phosphatase, aspartate aminotransferase, alanine aminotransferase, bilirubin, blood creatinine, total cholesterol, triglycerides, fasting glucose, total protein, globulin, calcium, sorbitol dehydrogenase, chloride, inorganic phosphorus, lipoprotein—low and high density, urea nitrogen, magnesium, potassium, and sodium. Activated partial thromboplastin time and prothrombin time were also evaluated, in addition to triiodothyronine, thyroxine, and thyroxine stimulating hormone. Urine was collected overnight for analysis of bilirubin, blood, color, clarity, quality, glucose, ketones, sediment (microscopic), pH, protein (total), specific gravity, volume, and urobilinogen. At euthanasia, vaginal smears were collected from the females to assess the stage of estrus. Following euthanasia, all animals underwent a complete necropsy, which included examination of the external surface of the body, the musculoskeletal system, and the contents of the cranial, thoracic, abdominal, and pelvic cavities, and all gross lesions were recorded. A select set of tissues were weighed wet, and included adrenals, brain, epididymides, heart, kidneys, liver, testes, ovaries with oviducts, spleen, thymus, and uterus. Paired organs were weighed together. The following list of tissues and organs were preserved in 10% neutral buffered formalin for histopathological evaluation: prostate and seminal vesicles, adrenals, aorta, femur, bone marrow from femur and sternum, brain, cecum, cervix, colon, duodenum, esophagus, Harderian gland, heart, ileum with Peyer’s patches, jejunum, kidneys, larynx, liver, lungs, mandibular and mesenteric lymph nodes, mammary gland, nasal turbinates, nose, ovaries, oviducts, pancreas, parathyroid, sciatic nerve, pharynx, pituitary gland, rectum, salivary glands, skeletal muscle, skin, spinal cord, spleen, sternum, stomach, thymus, thyroid, trachea, urinary bladder, uterus, vagina, and all gross lesions. The eyes, epididymides, optic nerve, and testes were preserved in modified Davidson’s fixative and then stored in 10% neutral buffered formalin, with the exception of the eyes and optic nerve, which were preserved in ethanol. Histological examinations were performed on all the preserved tissues and organs from the animals in the control and high-dose groups. The epididymis and testes from one low-dose male were prepared and examined microscopically due to abnormalities noted. The tissues were prepared by trimming the tissues, which were then processed, embedded in paraffin, sectioned, stained with hematoxylin and eosin, and then examined with light microscopy. Means and standard deviations were calculated for all quantitative data. In-life endpoints that were identified as multiple measurements of continuous data over time and treatment and control groups were compared using one-way analysis of variance (ANOVA), testing the effects of both time and treatment, with methods accounting for repeated measures in one independent variable (time) [23]. Significant interactions observed between treatment and time as well as main effects were further analyzed by a post hoc multiple comparisons test (e.g., Dunnett’s test) [24,25] of the individual treated groups to the control. All endpoints with single measurements of continuous data within groups, when warranted by sufficient group sizes, were evaluated for homogeneity of variances [26] and normality. Where homogeneous variances and normal distribution were observed, treated and control groups were compared using one-way ANOVA. When one-way ANOVA was significant, comparison of the treated groups to the control was performed with a multiple comparisons test (e.g., Dunnett’s test) [24,25]. Where variance was considered significantly different, groups were compared using a non-parametric method (e.g., Kruskal–Wallis non-parametric ANOVA) [27]. When non-parametric ANOVA was significant, a comparison of treated groups to the control was performed (e.g., Dunn’s test) [28]. For clinical pathology, Bartlett’s test for homogeneity and Shapiro–Wilk test for normality was done initially, and if no significance was seen, a one-way ANOVA was done, followed by Dunnett’s test. If the preliminary test was significant, log transformations of the data were done to achieve normality, and variance homogeneity was used. If the log transformation failed, a non-parametric method (e.g., Kruskal–Wallis non-parametric ANOVA) was used. When non-parametric ANOVA was significant, a comparison of treated groups to control was performed (e.g., Dunn’s test) [28]. Significance was judged at a probability value of *p* < 0.05. Male and female rats were evaluated separately.

## 3. Results

### 3.1. Bacterial Reverse Mutation (Ames) Assay

There were no treatment or concentration-related increases in the number of revertant colonies in any of the strains tested, both in the presence and absence of metabolic activation with either the plate incorporation or pre-incubation method (Appendix A). There was no precipitation or signs of toxicity observed in any strains at any dose levels. Some contamination was reported on several plates, but for all strains at least eight non-toxic dose levels without precipitate were evaluated. The mean revertant colony counts for each strain tested with the vehicle were close to or within the expected range based on historical controls and/or published values. The positive control articles caused expected substantial increases in the revertant colony counts.

### 3.2. Acute Single Dose Oral Toxicity Study

No mortalities were reported during the study, and bodyweights in the treated animals were comparable with the concurrent controls. No gross signs of toxicity or abnormal behaviors were reported during the 14-day observation period, and no macroscopic abnormalities were reported in any animals when necropsied at the end of the study. Under the conditions of this study, the acute oral lethal dose (LD_50_) was determined to be greater than 5000 mg/kg bw in female Sprague–Dawley rats.

### 3.3. Seven-Day Oral Range Finding Study

No mortalities or abnormal clinical observations were reported during the study other than the presence of black ocular discharge from the right eye of one female in the control group. Mean weekly bodyweight and daily bodyweight gains in the salidroside-treated males were comparable to concurrent controls, with the exception of a significant decrease in mean daily bodyweight gain in the mid-dose group on days 7 and 8 (Appendix A). This change was not considered to be toxicologically significant, as it was not dose-dependent. No significant changes were reported in the treated females. The mean daily food consumption in treated animals was comparable to concurrent controls, with the exception of a significant increase in the low- and mid-dose females on days 7 to 8 (Appendix A). These findings were not dose-dependent and not considered to be toxicologically relevant. No abnormal macroscopic observations were reported in any of the study animals.

### 3.4. Twenty-Eight-Day Repeat Dose Study

The salidroside was determined to be stable during the study, and analysis of the dose formulations considered them to meet the target concentrations at all dose levels on days 2, 10, and 28. No mortalities or salidroside-related abnormal clinical observations were reported during the course of the study. Incidental observations during the study included alopecia, moderate swelling in one paw, and abnormal gait in three control females. No salidroside-related abnormalities were reported during the FOB or motor activity analysis. Incidental findings in the FOB included a lack of tail-pinch response (males in the control and mid- and high-dose groups), and in females, hair loss (control), impaired gait and locomotion (control), and lack of tail-pinch response (control and low- and high-dose). The mean weekly bodyweights and daily bodyweight gain for all male and females in the salidroside groups were comparable to those of the controls throughout the study (Table 1 and Table 2).

Mean daily food consumption in all treated groups was also comparable to the concurrent controls, with the exception of significant decreases in mean daily food consumption in the mid-dose females on study days 1–8 (Table 3). These were only seen in one sex, not dose-dependent, and were not related to salidroside dosing.

Evaluation of the hematology, coagulation, clinical chemistry, thyroid hormones, and urinalysis parameters showed that repeat exposure to salidroside over 28-days did not induce any test-article-related changes (Table 4, Table 5 and Table 6). There were increased lymphocytes in the high-dose males, which were considered to be toxicologically insignificant as no signs of inflammation were evident in any other parameters evaluated.

The elevated BUN in the high-dose females was not considered to be toxicologically relevant based on the creatinine and urinalysis values. The increased protein values in the mid-dose females were considered to be toxicologically insignificant as there was no correlating abnormal histopathology in the kidneys. Decreased TSH was reported in mid- and high-dose males, increased T4 in the mid- and high-dose males, and increased T3 in both males and females. These changes were considered to be toxicologically insignificant due to the sporadic nature and that no abnormalities were reported in the thyroid gland. The stage of estrus was evaluated in all females at the end of the study to allow for histopathological evaluation of the female reproductive organs. There were no macroscopic alterations related to salidroside exposure found in any study animals; however, some incidental changes were reported. These included unilateral testicular atrophy in one low- and two high-dose males and a fluid-filled uterus in a high-dose female. Testicular atrophy is a common incidental finding in laboratory rats, and a fluid-filled uterus is related to the estrus cycle in females. All microscopic abnormalities reported were common, incidental changes frequently reported in laboratory rats [29]. No treatment-related changes in organ weights and ratios were reported; however, incidental findings included a statistically significant increase in the mean relative epididymides to brain weights in the high-dose males and significant increases in mean relative kidney to bodyweight and uterus to bodyweight in the high-dose females. These were considered to be incidental or due to biological variation as no abnormalities were reported in any organs during microscopic evaluation (Table 7 and Table 8).

## 4. Discussion

Extract from *Rhodiola rosea* contains a complex mixture of phytochemicals and is commonly used in Asia and in Europe as a supplement while gaining popularity in global markets. However, variation in constituents present in Rhodiola products is a cause for concern. To meet consumer demand, one possible method is to use nature-identical versions of key constituents created using synthetic biology. Synthetic biology approaches avoid the ecological implications of overharvesting endangered plants, circumvent low-yielding extraction methods, and reduce risk of adulteration of supplement products with other material [30].

The salidroside tested in this study is manufactured using a bioengineered *E. coli* strain in a rigorous manufacturing process subject to stringent quality controls. The engineered *E. coli* strain is cultured in a proprietary fermentation medium and allowed to ferment until a suitable level of the target ingredient (salidroside) is present. The subsequent processing steps kill any remaining bacteria and purify the chemical of interest. This process is similar to other common medicines, such as biopharmaceuticals, including insulin [31], or feed additives to replace antibiotics for livestock [32], and is expected to be safe. Based on the results of the in vitro and in vivo studies reported here, which include an in vitro bacterial reverse mutation assay as well as in vivo toxicity studies in male and female Sprague–Dawley rats, the bioengineered salidroside is not genotoxic and is a viable alternative to salidroside produced from harvested Rhodiola plants.

## 5. Conclusions

Synthetic biology provides the opportunity to access potent bioactive molecules that are otherwise inaccessible at scale. These limitations, which include abundance in nature, economics, and ethics behind sourcing raw material, can be circumvented through bioproduction of these molecules. Through strain engineering, synthetic biology offers a solution to obtaining these useful and therapeutic molecules. However, it is important to verify that molecules that are produced through synthetic biology and the overall bioproduction process are active and nontoxic, similar to the nature-derived versions of these molecules.

Based on the results of the Ames assay conducted, the bioengineered salidroside is not genotoxic in the bacterial mutation test. In addition, the results of the 28-day repeat dose oral toxicity study demonstrate that the no-observed-adverse-effect level (NOAEL) for the bioengineered salidroside is at least/greater than 2000 mg/kg bw/day in male and female Sprague–Dawley rats.

These results underscore the safety of a bioengineered salidroside product, even at high doses. Due to the apparent lack of any toxic effects, this bioengineered salidroside can be applied as an ingredient in dietary supplements, as a food ingredient, and potentially as a pharmaceutical therapeutic agent. Further research will focus on fully understanding the potential applications of bioengineered salidroside.

## Figures and Tables

**Table 1 nutrients-14-02330-t001:** Mean bodyweights for the 28-day repeat dose oral toxicity study with salidroside.

Day(s) Relative to Start Date	Control	500 mg/kg bw/day	1000 mg/kg bw/day	2000 mg/kg bw/day
Males (g)				
1	258.7 ± 20.6	255.7 ± 17.6	256.8 ± 16.9	256.1 ± 18.6
8	308.3 ± 17.7	303.9 ± 14.8	303.1 ± 19.8	309.7 ± 28.6
15	351.3 ± 20.9	343.2 ± 16.3	345.2 ± 23.8	347.1 ± 29.9
22	390.3 ± 22.1	383.3 ± 20.7	381.7 ± 27.1	387.6 ± 36.2
29	416.8 ± 23.8	411.3 ± 19.6	408.8 ± 29.6	417.8 ± 42.2
30 *	395.0 ± 20.1	384.9 ± 18.7	384.5 ± 30.1	391.3 ± 45.4
Females (g)				
1	218.8 ± 15.7	218.7 ± 15.1	219.2 ± 14.6	218.3 ± 14.7
8	239.6 ± 27.3	236.6 ± 13.0	230.7 ± 16.6	232.9 ± 17.4
15	253.6 ± 24.7	251.6 ± 17.5	245.1 ± 17.0	245.7 ± 18.1
22	270.2 ± 24.7	265.5 ± 16.6	262.2 ± 21.5	265.8 ± 19.1
29	278.0 ± 27.4	282.4 ± 22.5	275.3 ± 24.7	274.6 ± 21.5
31 *	261.8 ± 24.3	260.0 ± 17.0	256.6 ± 21.7	255.1 ± 19.1

* Fasted weight; n = 10/sex/group; mean ± standard deviation; bw = bodyweight; g = grams; kg = kilograms; mg = milligrams.

**Table 2 nutrients-14-02330-t002:** Mean daily bodyweight gain for the 28-day repeat dose oral toxicity study with salidroside.

Day(s) Relative to Start Date	Control	500 mg/kg bw/day	1000 mg/kg bw/day	2000 mg/kg bw/day
Males (g)				
1–8	7.09 ± 1.23	6.89 ± 1.57	6.61 ± 1.04	7.65 ± 1.90
8–15	6.14 ± 0.77	5.61 ± 0.92	6.01 ± 0.86	5.34 ± 1.02
15–22	5.57 ± 0.88	5.73 ± 1.13	5.21 ± 0.67	5.79 ± 1.33
22–29	3.79 ± 1.27	4.00 ± 0.39	3.87 ± 0.78	4.31 ± 1.13
1–29	5.65 ± 0.54	5.56 ± 0.56	5.43 ± 0.61	5.78 ± 0.94
Marginal	5.65 ± 1.43	5.56 ± 1.34	5.43 ± 1.21	5.78 ± 1.66
Females (g)				
1–8	2.97 ± 1.87	2.56 ± 0.72	1.64 ± 0.60	2.09 ± 1.29
8–15	2.00 ± 2.13	2.14 ± 1.12	2.06 ± 0.77	1.83 ± 0.92
15–22	2.37 ± 1.19	1.99 ± 1.49	2.44 ± 1.27	2.87 ± 0.86
22–29	1.11 ± 1.12	2.41 ± 1.51	1.87 ± 0.92	1.26 ± 0.66
1–29	2.11 ± 0.59	2.28 ± 0.32	2.00 ± 0.42	2.01 ± 0.36
Marginal	2.11 ± 1.55	2.27 ± 1.10	2.00 ± 0.86	2.01 ± 0.99

n = 10/sex/group; mean ± standard deviation; bw = bodyweight; g = grams; kg = kilograms; mg = milligrams.

**Table 3 nutrients-14-02330-t003:** Mean daily food consumption for the 28-day repeat dose oral toxicity study with salidroside.

Day(s) Relative to Start Date	Control	500 mg/kg bw/day	1000 mg/kg bw/day	2000 mg/kg bw/day
Males (g)				
1–8	25.13 ± 0.85	23.96 ± 0.94	24.04 ± 1.10	24.80 ± 2.31
8–15	25.01 ± 1.27	24.57 ± 0.66	24.14 ± 1.11	24.46 ± 1.98
15–22	26.19 ± 1.30	26.24 ± 1.15	25.49 ± 1.42	25.70 ± 2.41
22–29	25.47 ± 1.10	25.50 ± 0.74	24.80 ± 1.20	25.83 ± 2.75
1–29	25.45 ± 1.10	25.07 ± 0.76	24.62 ± 1.14	25.20 ± 2.33
Females (g)				
1–8	18.64 ± 1.78	18.63 ± 0.77	17.67 ± 0.18 *	18.13 ± 0.73
8–15	18.30 ± 0.82	18.29 ± 1.24	18.04 ± 0.42	18.30 ± 0.83
15–22	18.33 ± 1.46	18.87 ± 1.16	18.90 ± 0.79	19.10 ± 1.17
22–29	18.04 ± 0.91	19.27 ± 1.67	18.67 ± 0.65	18.36 ± 1.32
1–29	18.33 ± 0.99	18.76 ± 1.11	18.32 ± 0.38	18.47 ± 0.93

* Anova and Dunnett = *p* < 0.05; n = 10/sex/group; mean ± standard deviation; bw = bodyweight; g = grams; kg = kilograms; mg = milligrams.

**Table 4 nutrients-14-02330-t004:** Hematology and coagulation data for the 28-day repeat dose oral toxicity study with salidroside.

Parameter	Control	500 mg/kg bw/day	1000 mg/kg bw/day	2000 mg/kg bw/day
Males (30 days relative to start date)				
RBC (10^6^/µL)	8.404 ± 0.252	8.594 ± 0.321	8.299 ± 0.419	8.486 ± 0.242
HGB (g/dL)	15.20 ± 1.80	15.98 ± 0.52	15.67 ± 0.64	15.78 ± 0.33
HCT (%)	49.04 ± 1.92	50.38 ± 2.30	48.93 ± 2.20	49.93 ± 1.33
MCV (fL)	58.37 ± 2.24	58.62 ± 1.76	58.98 ± 0.88	58.88 ± 1.55
MCH (pg)	18.10 ± 1.98	18.60 ± 0.44	18.93 ± 0.49	18.61 ± 0.53
MCHC (g/dL)	30.94 ± 2.98	31.74 ± 0.59	32.09 ± 0.46	31.60 ± 0.39
ARET (×10^3^/µL)	258.400 ± 40.392	242.360 ± 25.541	237.830 ± 19.103	246.720 ± 51.160
PLT (×10^3^/µL)	1105.40 ± 134.89	1013.40 ± 169.13	985.40 ± 110.05	1126.40 ± 194.71
WBC (×10^3^/µL)	9.548 ± 2.204	9.691 ± 1.804	8.888 ± 2.097	11.437 ± 1.691
ANEU (×10^3^/µL)	1.749 ± 0.595	1.596 ± 0.432	1.560 ± 0.347	1.610 ± 0.371
ALYM (×10^3^/µL)	7.208 ± 1.877	7.487 ± 1.314	6.780 ± 1.866	9.147 ± 1.517 **
AMON (×10^3^/µL)	0.286 ± 0.114	0.330 ± 0.097	0.296 ± 0.099	0.341 ± 0.097
AEOS (×10^3^/µL)	0.129 ± 0.043	0.097 ± 0.027	0.090 ± 0.029	0.135 ± 0.059
ALUC (×10^3^/µL)	0.048 ± 0.019	0.062 ± 0.029	0.049 ± 0.028	0.079 ± 0.032 *
ABAS (×10^3^/µL)	0.126 ± 0.063	0.120 ± 0.058	0.112 ± 0.050	0.125 ± 0.063
RDW (%)	13.29 ± 0.52	13.02 ± 0.54	12.79 ± 0.46	13.01 ± 0.64
APTT (s)	16.82 ± 3.15	15.88 ± 1.46	17.18 ± 1.83	16.05 ± 3.19
PT (s)	9.35 ± 0.29	9.30 ± 0.15	9.38 ± 0.24	9.43 ± 0.33
Females (31 days relative to start date)				
RBC (10^6^/µL)	8.022 ± 0.276	8.199 ± 0.363	7.905 ± 0.354	8.112 ± 0.227
HGB (g/dL)	14.89 ± 0.49	15.33 ± 0.58	14.96 ± 0.58	14.94 ± 0.47
HCT (%)	44.62 ± 1.29	46.50 ± 1.96	45.39 ± 1.80	45.29 ± 1.63
MCV (fL)	55.66 ± 1.06	56.74 ± 1.63	57.44 ± 2.23	55.84 ± 1.69
MCH (pg)	18.59 ± 0.32	18.71 ± 0.57	18.93 ± 0.73	18.43 ± 0.52
MCHC (g/dL)	33.41 ± 0.26	32.98 ± 0.36 ***	32.94 ± 0.38 ***	32.99 ± 0.41 ***
ARET (×10^3^/µL)	186.480 ± 34.025	193.580 ± 46.509	215.090 ± 53.276	195.160 ± 32.392
PLT (×10^3^/µL)	1147.30 ± 172.22	1070.80 ± 131.26	1069.90 ± 77.77	1145.50 ± 103.41
WBC (×10^3^/µL)	7.590 ± 1.386	8.017 ± 1.404	6.972 ± 2.102	7.344 ± 2.352
ANEU (×10^3^/µL)	1.407 ± 0.322	1.523 ± 0.585	1.233 ± 0.357	1.377 ± 0.492
ALYM (×10^3^/µL)	5.690 ± 1.357	6.035 ± 1.171	5.364 ± 1.917	5.580 ± 2.138
AMON (×10^3^/µL)	0.246 ± 0.145	0.203 ± 0.100	0.165 ± 0.102	0.157 ± 0.092
AEOS (×10^3^/µL)	0.127 ± 0.044	0.088 ± 0.043	0.087 ± 0.035	0.097 ± 0.037
ALUC (×10^3^/µL)	0.053 ± 0.021	0.046 ± 0.018	0.040 ± 0.020	0.049 ± 0.034
ABAS (×10^3^/µL)	0.072 ± 0.033	0.124 ± 0.036 ***	0.082 ± 0.055	0.084 ± 0.034
RDW (%)	11.55 ± 0.22	11.63 ± 0.50	11.79 ± 0.32	11.57 ± 0.37
APTT (s)	14.59 ± 1.37	14.79 ± 2.55	15.70 ± 3.08	14.97 ± 3.36
PT (s)	8.75 ± 0.20	9.03 ± 0.22	9.05 ± 0.59	8.90 ± 0.31

n = 10/sex/group. Data are presented as mean ± standard deviation. * Anova and Dunnett (Log) = *p* < 0.05; ** Anova and Dunnett = *p* < 0.05; *** Anova and Dunnett = *p* < 0.05. APTT = activated partial thromboplastin time; bw = bodyweight; dL = deciliter; AEOS = absolute eosinophils; fL = femtoliters; g = grams; HCT = hematocrit; HGB = hemoglobin; kg = kilogram; L = liters; ALUC = absolute large unstained cell; ALYM = absolute lymphocytes; AMON = absolute monocytes; ANEU = absolute neutrophils; ARET = absolute reticulocytes; MCH = mean corpuscular hemoglobin; MCHC = mean corpuscular hemoglobin concentration; MCV = mean corpuscular volume; mg = milligrams; pg = picograms; PT = prothrombin time; RBC = erythrocytes; RDW = red cell distribution width; s = seconds; TB = thrombocytes/platelets; WBC = white blood cells (leukocytes).

**Table 5 nutrients-14-02330-t005:** Clinical chemistry and thyroid hormone data for the 28-day repeat dose oral toxicity study with salidroside.

Parameter	Control	500 mg/kg bw/day	1000 mg/kg bw/day	2000 mg/kg bw/day
Males (30 days relative to start date)				
AST (U/L)	96.1 ± 17.2	105.9 ± 18.8	101.8 ± 15.7	103.7 ± 31.2
ALT (U/L)	30.4 ± 3.9	32.2 ± 6.1	32.9 ± 6.0	28.1 ± 4.5
ALKP (U/L)	141.5 ± 15.1	147.7 ± 26.8	120.9 ± 32.2	136.0 ± 32.1
BILI (µmol/L)	0.060 ± 0.013	0.057 ± 0.020	0.060 ± 0.021	0.064 ± 0.028
BUN (mmol/L)	14.5 ± 1.8	15.8 ± 1.7	14.2 ± 2.3	17.2 ± 1.7 ***
CREA (µmol/L)	0.202 ± 0.025	0.206 ± 0.029	0.194 ± 0.028	0.186 ± 0.024
CHOL (mmol/L)	58.0 ± 9.5	57.6 ± 9.1	65.3 ± 16.6	58.8 ± 11.4
LDL (mmol/L)	0.260 ± 0.084	0.250 ± 0.085	0.260 ± 0.117	0.190 ± 0.074
HDL (mmol/L)	1.010 ± 0.173	0.990 ± 0.145	1.100 ± 0.313	0.990 ± 0.208
GLUC (mmol/L)	158.2 ± 35.6	167.9 ± 34.2	180.7 ± 43.5	207.2 ± 56.1
TP (g/L)	5.96 ± 0.19	5.88 ± 0.22	5.94 ± 0.35	5.95 ± 0.35
ALB (g/L)	4.08 ± 0.14	4.01 ± 0.17	4.00 ± 0.24	4.08 ± 0.20
GLOB (g/L)	1.88 ± 0.12	1.87 ± 0.13	1.94 ± 0.16	1.87 ± 0.23
Ca (mg/dL)	11.49 ± 0.29	11.67 ± 0.48	12.04 ± 0.91	12.49 ± 1.24 *
Mg (mmol/L)	1.261 ± 0.147	1.252 ± 0.201	1.234 ± 0.241	1.217 ± 0.158
Na (mmol/L)	143.60 ± 1.35	144.60 ± 1.43	145.50 ± 1.72 **	145.70 ± 1.06 ***
K (mmol/L)	8.191 ± 1.137	7.303 ± 1.176	7.020 ± 1.513	7.157 ± 1.026
Cl (mmol/L)	104.42 ± 1.00	103.93 ± 1.09	104.01 ± 1.47	104.75 ± 1.51
IPHS (mg/dL)	11.26 ± 0.73	11.00 ± 1.19	10.96 ± 1.32	11.41 ± 0.95
SDH (U/L)	12.93 ± 8.58	11.66 ± 4.66	14.64 ± 8.97	11.25 ± 3.45
TRIG (mg/dL)	63.6 ± 22.0	53.9 ± 16.5	76.2 ± 42.0	65.0 ± 20.4
TSH (ng/mL)	5.0047 ± 0.8580	4.5183 ± 0.3315	4.0476 ± 0.5130 ^#^	3.7745 ± 0.4390 ^##^
TT3 (ng/mL)	1.1256 ± 0.0294	1.2533 ± 0.0746 ^####^	1.3509 ± 0.0942 ^####^	1.2416 ± 0.0817 ^###^
TT4 (ng/mL)	40.9020 ± 2.9969	43.0894 ± 2.3707	46.7989 ± 2.9682 ^####^	44.5315 ± 2.1342 ^###^
Females (31 days relative to start date)				
AST (U/L)	87.9 ± 10.1	81.2 ± 11.8	98.7 ± 26.3	100.2 ± 26.2
ALT (U/L)	24.0 ± 4.4	26.6 ± 7.4	29.1 ± 12.5	33.9 ± 13.5
ALKP (U/L)	68.4 ± 14.9	70.6 ± 18.4	65.0 ± 16.8	61.3 ± 16.8
BILI (µmol/L)	0.066 ± 0.013	0.070 ± 0.022	0.074 ± 0.019	0.067 ± 0.013
BUN (mmol/L)	17.1 ± 2.2	17.5 ± 3.2	18.5 ± 2.6	19.7 ± 3.3
CREA (µmol/L)	0.233 ± 0.037	0.243 ± 0.029	0.231 ± 0.040	0.242 ± 0.035
CHOL (mmol/L)	76.3 ± 14.9	70.6 ± 16.1	77.2 ± 16.8	79.1 ± 26.3
LDL (mmol/L)	0.268 ± 0.071	0.223 ± 0.078	0.231 ± 0.093	0.234 ± 0.115
HDL (mmol/L)	1.472 ± 0.282	1.375 ± 0.289	1.531 ± 0.317	1.601 ± 0.467
GLUC (mmol/L)	164.5 ± 39.4	155.5 ± 47.4	160.3 ± 35.5	184.4 ± 58.6
TP (g/L)	6.23 ± 0.50	6.35 ± 0.28	6.31 ± 0.34	6.75 ± 0.40 *
ALB (g/L)	4.43 ± 0.49	4.63 ± 0.22	4.68 ± 0.33	4.99 ± 0.33 ***
GLOB (g/L)	1.80 ± 0.29	1.72 ± 0.12	1.63 ± 0.16	1.76 ± 0.22
Ca (mg/dL)	11.48 ± 0.50	11.80 ± 0.59	12.00 ± 0.70	12.79 ± 0.98 ****
Mg (mmol/L)	1.182 ± 0.162	1.316 ± 0.117	1.228 ± 0.200	1.283 ± 0.139
Na (mmol/L)	136.40 ± 1.35	137.40 ± 1.07	140.80 ± 1.81 ****	141.50 ± 1.78 ****
K (mmol/L)	7.088 ± 1.215	8.514 ± 1.473	7.437 ± 2.325	7.877 ± 1.647
Cl (mmol/L)	100.39 ± 1.34	101.53 ± 1.68	103.25 ± 1.80 ***	102.81 ± 2.03 ***
IPHS (mg/dL)	9.53 ± 1.12	10.62 ± 1.22	10.43 ± 1.72	10.02 ± 1.18
SDH (U/L)	9.17 ± 3.43	11.63 ± 2.08	14.95 ± 9.74	14.86 ± 9.82
TRIG (mg/dL)	37.8 ± 10.4	39.2 ± 8.4	41.8 ± 14.7	47.0 ± 12.8
TSH (ng/mL)	3.9470 ± 0.1790	4.3606 ± 0.3872	3.7501 ± 0.2724	4.0032 ± 0.5661
TT3 (ng/mL)	1.2243 ± 0.0981	1.3705 ± 0.2550 *	1.9454 ± 0.2898 ^##^	1.8470 ± 0.4490 ^##^
TT4 (ng/mL)	40.5269 ± 3.6993	38.3678 ± 2.4368	40.5570 ± 4.5737	38.5460 ± 3.9566

n = 10/sex/group. Data are presented as mean ± standard deviation (SD). * Anova and Dunnett (Rank) = *p* <0.05; ** Anova and Dunnett = *p* < 0.05; *** Anova and Dunnett = *p* < 0.01; **** Anova and Dunnett = *p* < 0.001; ^#^ Anova and Dunnett (Rank) = *p* <0.01; ^##^ Anova and Dunnett (Rank) = *p* < 0.001; ^###^ Anova and Dunnett (Log) = *p* < 0.01; ^####^ Anova and Dunnett (Log) = *p* < 0.001 ALB = albumin; ALKP = alkaline phosphatase; ALT = alanine aminotransferase; AST = aspartate aminotransferase; BILI = total bilirubin; BUN = urea nitrogen; Ca = calcium; CHOL = cholesterol; Cl = chloride; CREA = creatinine; g = grams; GLOB = globulin; GLUC = glucose; HDL = high density lipoprotein cholesterol; K = potassium; L = liters; LDL = low density lipoprotein cholesterol; Mg = magnesium; mg = milligrams; ml = milliliter; mmol = millimoles; Na = sodium; ng = nanograms; IPHS = inorganic phosphorus; SDH = sorbital dehydrogenase; TP = total protein; TRIG = triglycerides; TSH = thyroid stimulating hormone; TT3 = total triiodothyronine; TT4 = thyroxine; U = units; µmol = micromoles.

**Table 6 nutrients-14-02330-t006:** Urinalysis data for the 28-day repeat dose oral toxicity study with salidroside.

Parameter	Control	500 mg/kg bw/day	1000 mg/kg bw/day	2000 mg/kg bw/day
Males				
Urine Volume (ml)	16.30 ± 10.70	8.80 ± 6.81	10.80 ± 5.88	12.45 ± 8.16
pH	7.05 ± 0.69	6.20 ± 0.35 ^##^	6.10 ± 0.21 ^##^	6.00 ± 0.41 ^##^
Glucose (mg/dL)	0	0 ^x^	0 ^x^	0 ^x^
Ketone (mmol/L)	8.0 ± 4.8	14.0 ± 3.2 ^#^	10.0 ± 5.3	5.0 ± 0.0
Protein (mg/dL)	10.5 ± 10.1	16.5 ± 11.1	10.5 ± 12.3	13.5 ± 11.1
Specific Gravity	1.0150 ± 0.0082	1.0250 ± 0.0075 ^#^	1.0255 ± 0.0050 ^#^	1.0230 ± 0.0059 ***
Urobilinogen	0.20 ± 0.00	0.20 ± 0.00 ^x^	0.20 ± 0.00 ^x^	0.20 ± 0.00 ^x^
Females				
Urine Volume (ml)	4.65 ± 3.59	5.50 ± 6.16	3.30 ± 4.39	3.45 ± 2.31
pH	6.50 ± 0.62	6.17 ± 0.61	5.90 ± 0.77	5.40 ± 0.52 **
Glucose (mg/dL)	0	0 ^x^	0 ^x^	0 ^x^
Ketone (mmol/L)	0.5 ± 1.6	0.0 ± 0.0	0.0 ± 0.0	0.5 ± 1.6
Protein (mg/dL)	12.0 ± 13.8	13.3 ± 13.9	226.5 ± 623.8	61.0 ± 87.3
Specific Gravity	1.0220 ± 0.0071	1.0244 ± 0.0068	1.0265 ± 0.0075	1.0300 ± 0.0000 ***
Urobilinogen	0.20 ± 0.00	0.20 ± 0.00 ^x^	0.20 ± 0.00 ^x^	0.20 ± 0.00 ^x^

n = 10/sex/group. Data presented as mean ± standard deviation. Mg—milligram; dL—deciliter; mL—milliliter; mmol—millimole; L—liter. ^x^ = not appropriate for statistics; ** Anova and Dunnett = *p* < 0.01; *** Anova and Dunnett = *p* < 0.05; ^#^ Anova and Dunnett (Rank) = *p* < 0.01; ^##^ Anova and Dunnett (Rank) = *p* < 0.001.

**Table 7 nutrients-14-02330-t007:** Organ-to-bodyweight ratio.

Examined Organ	Control	500 mg/kg bw/day	1000 mg/kg bw/day	2000 mg/kg bw/day
Males				
Adrenal	0.1762 ± 0.0382	0.1634 ± 0.0304	0.1517 ± 0.0190	0.1598 ± 0.0237
Brain	5.349 ± 0.340	5.560 ± 0.331	5.642 ± 0.530	5.572 ± 0.587
Epididymides	3.0120 ± 0.3136	2.8484 ± 0.2118	2.8634 ± 0.2975	2.7076 ± 0.4264
Heart	3.209 ± 0.132	3.294 ± 0.174	3.332 ± 0.339	3.536 ± 0.418
Kidneys	6.876 ± 0.363	6.907 ± 0.581	6.630 ± 0.449	7.198 ± 0.709
Liver	30.136 ± 3.144	29.114 ± 2.377	28.251 ± 2.635	29.747 ± 2.674
Spleen	1.816 ± 0.346	1.910 ± 0.115	1.980 ± 0.131	1.924 ± 0.253
Testes	8.830 ± 0.767	8.410 ± 1.084	8.513 ± 1.189	7.954 ± 1.430
Thymus	1.2051 ± 0.3112	1.2067 ± 0.2922	1.0406 ± 0.2464	1.1377 ± 0.3418
Females				
Adrenal	0.2960 ± 0.0609	0.2918 ± 0.0659	0.2623 ± 0.0645	0.2954 ± 0.0461
Brain	7.974 ± 0.829	7.879 ± 0.589	8.207 ± 0.852	8.316 ± 0.551
Heart	3.447 ± 0.157	3.398 ± 0.185	3.487 ± 0.194	3.427 ± 0.230
Kidneys	7.146 ± 0.442	7.172 ± 0.424	7.479 ± 0.474	7.958 ± 0.685 **
Liver	31.482 ± 2.031	32.116 ± 2.625	32.139 ± 3.048	33.788 ± 2.547
Ovaries with oviducts	0.5372 ± 0.1020	0.5095 ± 0.1322	0.5038 ± 0.0484	0.5420 ± 0.0642
Spleen	2.142 ± 0.273	2.224 ± 0.347	2.111 ± 0.167	2.269 ± 0.496
Thymus	1.5372 ± 0.3758	1.8118 ± 0.3828	1.6991 ± 0.2574	1.6189 ± 0.3721
Uterus	1.966 ± 0.387	2.112 ± 0.607	2.409 ± 0.896	2.845 ± 0.910 *

n = 10/sex/group. Data presented as mean ± standard deviation. * Anova and Dunnett (Log) = *p* < 0.05; ** Anova and Dunnett = *p* < 0.01.

**Table 8 nutrients-14-02330-t008:** Organ-to-brain weight ratio.

Examined Organ	Control	500 mg/kg bw/day	1000 mg/kg bw/day	2000 mg/kg bw/day
Males				
Adrenal	0.0323 ± 0.0065	0.0296 ± 0.0063	0.0271 ± 0.0040	0.0290 ± 0.0052
Epididymides	0.5647 ± 0.0646	0.5134 ± 0.0427	0.5090 ± 0.0485	0.4884 ± 0.0775 *
Heart	0.602 ± 0.034	0.594 ± 0.049	0.592 ± 0.049	0.645 ± 0.137
Kidneys	1.290 ± 0.101	1.247 ± 0.144	1.182 ± 0.119	1.296 ± 0.100
Liver	5.639 ± 0.509	5.241 ± 0.397	5.051 ± 0.718	5.407 ± 0.865
Spleen	0.339 ± 0.060	0.345 ± 0.027	0.353 ± 0.033	0.350 ± 0.066
Testes	1.658 ± 0.189	1.514 ± 0.184	1.507 ± 0.121	1.434 ± 0.250
Thymus	0.2255 ± 0.0578	0.2189 ± 0.0580	0.1845 ± 0.0401	0.2054 ± 0.0617
Females				
Adrenal	0.0371 ± 0.0064	0.0371 ± 0.0088	0.0317 ± 0.0064	0.0357 ± 0.0061
Heart	0.435 ± 0.038	0.434 ± 0.042	0.429 ± 0.051	0.413 ± 0.030
Kidneys	0.904 ± 0.099	0.915 ± 0.093	0.918 ± 0.092	0.957 ± 0.047
Liver	3.983 ± 0.451	4.106 ± 0.540	3.968 ± 0.677	4.078 ± 0.403
Ovaries with oviducts	0.0678 ± 0.0137	0.0648 ± 0.0168	0.0620 ± 0.0093	0.0652 ± 0.0070
Spleen	0.270 ± 0.035	0.283 ± 0.046	0.260 ± 0.036	0.272 ± 0.055
Thymus	0.1948 ± 0.0507	0.2301 ± 0.0476	0.2099 ± 0.0419	0.1957 ± 0.0480
Uterus	0.251 ± 0.069	0.270 ± 0.085	0.301 ± 0.129	0.339 ± 0.097

n = 10/sex/group. Data presented as mean ± standard deviation. * Anova and Dunnett = *p* <0.05.

## Data Availability

The data presented in this study are available in Kasprzyk, P.G.; Vickery, C.; Ye, M.; Sewastianik, M.; Gong, W., Ding, S.; Dziwenka, M., Mozingo, A.; Valm, K.; Schachner, H.; Weng, JK. Safety of a sustainably produced, bioengineered, nature-identical salidroside compound. *Nutrients* 2022, *14*, 2330. https://doi.org/10.3390/nu14112330.

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
