# Peer review of "Safety of a Sustainably Produced, Bioengineered, Nature-Identical Salidroside Compound"

_nutrients, 2022, doi:10.3390/nu14112330_

Round 1

Reviewer 1 Report

Currently, due to the increased incidence of civilization diseases, there is an increased interest in bioactive substances that have a positive impact on human health, especially those that will reduce the adverse effects of stress on the human body. Salidroside is one of such compounds, characterized by adaptogenic activity - it helps to fight stress. It influences on mood improvement by increased production of endorphins. However, due to the fact that the plant from which this compound is obtained grows very slowly and due to climatic changes occurring around the world, it is right to look for new ways of obtaining pro-health compounds. One of them is genetic engineering using microorganisms to produce such compounds. In this publication the authors decided to demonstrate the safety of the “nature-identical” salidroside  by means of the Bacterial Reverse Mutation (Ames) assay  to evaluate  if a compound can change the genetic sequence in a selection of bacteria as an indication  of mutagenicity potential and the single dose toxicity study gives an indication of acute toxicity, and the dose were used is often well above the intended use. The whole experiment was prepared and performed properly and met all the requirements of scientific experiment.

The authors indicated that the bioengineered salidroside is not genotoxic in the bacterial mutation test and the results of the 28-day repeat dose oral toxicity study demonstrate that the no-observed-adverse-effect-level (NOAEL) for the bioengineered salidroside is at least/greater than 2000 mg/kg bw/day in male and  female Sprague-Dawley rats,

which in my opinion in very interesting way shows the great practical value of this research.

I think that with such an interesting experiment, minor corrections should be made in the summary and conclusion sections, i.e:
summary - too extensive information about reasons and lack of detailed results summary - too little information about the possibility of using this compound and the costly process of obtaining salidroside, and how it could be used as a supplement or food additive?

Additionally, I wonder if with this type of research it would be reasonable to determine the immunomodulatory properties of this compound?

Author Response

RESPONSE TO REVIEWER 1.

Thank you for taking the time to review the manuscript and provide valuable feedback. Please find our responses below.

  1. Summary too extensive regarding reasons and lack of detailed summary results and too little information about the possibility of using this compound and the costly process of obtaining salidroside, and how it could be used as a supplement or food additive?
    1. RESPONSE: The abstract and conclusion sections have been edited based on the suggestions.
  2. Additionally, I wonder if with this type of research, it would be reasonable to determine the immunomodulatory properties of this compound?
    1. RESPONSE: Thank you for this suggestion. The aim of the studies which were conducted were to demonstrate the safety of the salidroside. In these studies, efficacy was not evaluated, however we will consider these types of studies in the future.

Reviewer 2 Report

The paper submitted to Nutrients by Kasprzyk et al. is focused on the evaluation of safety of salidroside . In my opinion the paper is generally well written and all experiments are correctly performed. However, some minor corrections are needed.

1) section 2 describing materials and methods should be divided in specific subsections - in the present form it is hard to follow by the reader

2) more info on the source and purity of salidroside used for the research should be provided - please state what was the purity of the compound and how the authenticity was confirmed

Author Response

 RESPONSE TO REVIEWER 2.

Thank you for taking the time to review the manuscript and provide valuable feedback. Please find our responses below.

  1. Section 2 describing materials and methods should be divided in specific subsections - in the present form it is hard to follow by the reader.
    1. RESPONSE: Subsections have been added to the materials and methods section.
  2. More info on the source and purity of salidroside used for the research should be provided - please state what was the purity of the compound and how the authenticity was confirmed
    1. RESPONSE: The purity of the test material used is stated in the material and methods section as follows: The test article used in the studies was salidroside (white crystalline powder; 98.8 – 99.3% pure, CAS #10338-51-9) and was supplied by LandKind, a subsidiary of DoubleRainbow Biosciences Inc.”. The same batch of salidroside was used for all studies and the sentence quoted above has been corrected to state the purity as 99.3% rather than a range. A GLP compliant method validation study was conducted prior to the 28-day study which was then used to evaluate the salidroside dosing formulations in the 28-day study. A certificate of analysis was provided to the test facility as well.

Reviewer 3 Report

Philip G. Kasprzyk et al. conducted standard assays and studies to prove the safety of the “nature-identical” salidroside. Their results well support that the salidroside produced using bioengineered E. coli is a viable alternative to salidroside produced from harvested Rhodiola spp. Only minor concerns need to be addressed.

1. Since salidroside is manufactured via a patented sustainable bioengineering manufacturing process, it is necessary to cite this patent or related publications.

2. In Acute Single Dose Oral Toxicity Study, why did the authors conduct solely in female Sprague-Dawley rats?

3. Some test results are missing. For example, vaginal smears from the females to assess the stage of estrus.

4. In “3.1. Bacterial Reverse Mutation (Ames) Assay”, the colony pictures shall be presented.

5. The histopathological pictures of tissues and organs should be added.

Author Response

RESPONSE TO REVIEWER 3.

Thank you for taking the time to review the manuscript and provide valuable feedback. Please find our responses below.

  1. Since salidroside is manufactured via a patented sustainable bioengineering manufacturing process, it is necessary to cite this patent or related publications.
    1. RESPONSE: A citation for patent US20190264221A1 has been added to the manuscript.
  2. In “Acute Single Dose Oral Toxicity Study”, why did the authors conduct solely in female Sprague-Dawley rats?
    1. RESPONSE: This study was designed based on OECD Guidelines for the Testing of Chemicals, Test No. 425 which states that “Normally female rats are used. This is because literature surveys of conventional LD50 tests show that usually there is little difference in sensitivity between sexes, but in those cases where differences are observed, females are generally slightly more sensitive…..”. There was no indication that both sexes would be required and therefore, to reduce animal use, only one sex was chosen. The OECD guideline can be found here Test No. 425: Acute Oral Toxicity: Up-and-Down Procedure | OECD Guidelines for the Testing of Chemicals, Section 4 : Health Effects | OECD iLibrary (oecd-ilibrary.org).
  3. Some test results are missing. For example, vaginal smears from the females to assess the stage of estrus.
    1. RESPONSE: Edits have been added to the manuscript clarifying that the vaginal smears were taken only to assist the Pathologist in evaluating any histopathological findings in the female reproductive organs.
  4. In “3.1. Bacterial Reverse Mutation (Ames) Assay”, the colony pictures shall be presented.
    1. RESPONSE: Unfortunately, no photos were taken by the testing facility.
  5. The histopathological pictures of tissues and organs should be added.
    1. RESPONSE: Again, unfortunately no photos were taken by the testing facility.